# Analysis of the RNA-Dependent RNA Polymerase 1 (RDR1) Gene Family in Melon

**DOI:** 10.3390/plants11141795

**Published:** 2022-07-07

**Authors:** Diana Leibman, Ekaterina Pashkovsky, Yulia Shnaider, Meital Shtarkman, Victor Gaba, Amit Gal-On

**Affiliations:** Department of Plant Pathology and Weed Research, Agricultural Research Organization, The Volcani Center, 68 HaMaccabim Road, P.O. Box 15159, Rishon LeZion 7505101, Israel; diana@agri.gov.il (D.L.); katuhad@gmail.com (E.P.); yula_g7@hotmail.com (Y.S.); meital.shtarkman@mail.huji.ac.il (M.S.); vpgaba@volcani.agri.gov.il (V.G.)

**Keywords:** RNA-dependent RNA polymerase 1 (RDR1), melon, plant virus, cucumber mosaic virus, virus susceptibility

## Abstract

RNA-dependent RNA polymerase 1 (RDR1) plays a crucial defense role against plant viruses by secondary amplification of viral double-stranded RNA in the gene-silencing pathway. In this study, it was found that melon *(Cucumis melo)* encodes four RDR1 genes (*CmRDR1a, b, c1* and *c2)* similar to the *CsRDR1* gene family of cucumber (*C*. *sativus*). However, in contrast to cucumber, melon harbors a truncated *CmRDR1b* gene. In healthy plants, *CmRDR1a* was expressed, whereas the expression of *CmRDR1c1/c2* was not detected. *CmRDR1a* expression level increased 20-fold upon cucumber mosaic virus (CMV) infection and was not increased in melon plants infected with zucchini yellow mosaic virus (ZYMV), cucumber vein yellowing virus (CVYV) and cucumber green mottle mosaic virus (CGMMV). The expression of *CmRDR1c1/c2* genes was induced differentially by infection with viruses from different families: high levels of ~340-, 172- and 115-fold increases were induced by CMV, CVYV and CGMMV, respectively, and relatively low-level increases by potyvirus infection (4- to 6-fold). CMV mutants lacking the viral silencing suppressor 2b protein did not cause increased *CmRDR1c/c2* expression; knockout of *CmRDR1c1/c2* by CRISPR/Cas9 increased susceptibility to CMV but not to ZYMV. Therefore, it is suggested that the sensitivity of melon to viruses from different families is a result of the loss of function of *CmRDR1b*.

## 1. Introduction

Melon is an important vegetable crop and melon world production was 28 million tons on 1.1 million hectares in 2020 [1]. RNA viruses are very serious threats to melon production and early virus infection can lead to total loss [2]. Plant virus infection triggers the plant immune system (post-transcriptional silencing and RNA silencing) by detection of viral replication in the dsRNA form [3]. Endogenous RNA-dependent RNA polymerases (RDRs) generate dsRNA virus fragments as target molecules for gene silencing [4].

Plant RNA silencing is a cellular defense mechanism that regulates transcription or post-transcriptional gene expression in a sequence-specific manner by small interfering RNA (siRNA) molecules [5,6,7]. The silencing mechanism is associated with plant defense against viral infection and RDRs are core factors that initiate the biogenesis of viral-siRNAs [8].

In plants there are six different RDRs; however, RDR1 and RDR6 are the dominant enzymes that amplify single-stranded RNA viruses into aberrant dsRNA, which are digested by the host-encoded Dicer-like (DCL) DCL-4 and DCL-2 proteins into 21–22 nts virus-siRNA duplexes [9,10,11,12]. In addition, during RNA virus infection biogenesis of endogenous small-RNA is activated, dependent on RDR1 activity [13]. 

RDR1, as a single or duplicated gene occurs in all investigated plant species; however, in cucurbits there is a small RDR1 gene family [14,15,16]. The expression levels of RDR1, induced by DNA and RNA viruses [14,17], phytohormones [18], fungal [19] and insect attack [20], and can be controlled by the alternative oxidase (AOX)-associated defense pathway [21] and plant microRNAs [22]. It has been shown that RDRs play an important role in addition to virus defense: RDR1 is involved with responses to biotic, abiotic stress and defense against insect herbivores via dsRNA production that leads to rapid RNA degradation [23,24,25]. Novel RNA regulation via RDR1/2/6-mediated biosynthesis of antisense RNAs has been described as a response to abiotic stress [23].

The association of virus resistance/susceptibility with low/absent RDR1 expression was first described in *Nicotiana bentamiana*, a natural *RDR1* mutant (with an insertion of 72 nts) that is highly susceptible to a wide range of viruses [26]. Transgenic *N. bentamiana* transformed with RDR1 genes from a number of plant species exhibited increased resistance to virus infection [26,27]. Correspondingly, virus accumulation increased when *RDR1* was silenced in species such as Arabidopsis, tomato, tobacco and potato [17,21,28]. However, overexpression of tobacco *NtRDR1* in *N. bentamiana* increased virus susceptibility [29] and silencing of potato *RDR1* did not affect virus susceptibility [15]. 

RNA virus infection induced salicylic acid accumulation, which led to RDR1 induction [30]. Pretreatment with salicylic acid induced RDR1 expression and enhanced defense against RNA and DNA viruses in several species, whereas decrease of endogenous salicylic acid levels hindered defense against virus infection [31,32,33]. 

RDR1, known to regulate microRNA levels [24] in rice, plays a role in regulating important endogenous genes via mRNA-mediated DNA methylation, and is associated with the abiotic stress response [34]. In addition, regulation of RDR1 in rice depended on mir144, which was activated by rice stripe virus infection [22]. Notably, RDR1 mutant plants of different species showed no discernible altered phenotypes in growth and development. In contrast, mutants in the RDR6 gene display disturbed leaf development, as RDR6 is essential for tasiRNA biogenesis [13,34,35]. 

The discovery of the unique cucurbit *RDR1* gene family was based on data from the Cucurbit Genomics Database. In cucumber, four functional *CsRDR1* genes were identified (*CsRDR1a*, *CsRDR1b*, *CsRDR1c1* and *CsRDR1c2*) [14]. In healthy cucumber plants, *CsRDR1a* and *CsRDR1b* were expressed, whereas *CsRDR1c1* and *CsRDR1c2* expression was undetectable. However, virus infection (especially cucumber mosaic virus (CMV) and cucumber vein yellowing virus (CVYV)) dramatically increased *CsRDR1c1* and *CsRDR1c2* expression, whereas silencing of *CsRDR1c1+2* led to increased virus accumulation [14]. In addition, constitutive high levels of *CsRDR1b* expression were associated with broad virus resistance [14]. 

Melon (*Cucumis melo*) and cucumber (*C. sativus*) are close relatives and belong to the *Cucumis* genus (*Cucurbitaceae*). The genomes of melon and cucumber are highly conserved despite genome duplication in melon (ca. 454 Mbp in 12 chromosome pairs), whereas cucumber has 7 chromosome pairs with a genome size of 367 Mbp. Melon is known to be susceptible to infection by a range of viruses of different families such as zucchini yellow mosaic virus (ZYMV), papaya ringspot virus (PRSV-W), water melon mosaic virus (WMV) and cucumber vein yellowing virus (CVYV) (family *Potyviridae*), CMV (*Bromoviridae*) and cucumber green mottle mosaic virus (CGMMV; *Virgaviridae*). Classical breeding for virus resistance in melon has proved insufficient due to limited genetic resistance sources.

In the present study, we characterized the four melon genes of the *CmRDR1* family (*CmRDR1a*, *CmRDR1b*, *CmRDR1c1* and *CmRDR1c2*) and the differential gene expression responses of plants infected with different viruses. To help in our understanding of the role of *CmRDR1c1/c2* genes in the plant response to viral disease, these genes were knocked out using CRISPR/Cas9 technology.

## 2. Materials and Methods

### 2.1. Plants, Pathogens and Inoculations

*Cucumis melo* L. cvs. Arava and Vedrantais were used for *CmRDR* gene mapping and virus susceptibility analyses. In addition, in 10 melon genotypes (listed in Appendix A) the truncated RDR1b genes were mapped. Seeds were planted in a soil mixture and grown in a greenhouse in daylight at 25 °C. The viruses listed in Appendix A were used for virus accumulation analysis and evaluation of RDR gene expression, namely: cucumber mosaic virus CMV—Fny strain, CMV mutant lacking the 2b gene (CMV-Δ2b) [36], CGMMV, CVYV, ZYMV, PRSV-W and WMV. Squash (*Cucurbita pepo* L. cv. Zuccini) plants were a source of inocula for PRSV-W, WMV and ZYMV. Cucumber (*Cucumis sativus* L. cv. Bet Alfa) plants were inocula sources for CMV, CMV-Δ2b, CGMMV and CVYV. Melon seedlings at the cotyledon stage with a small true leaf (about 3–5 days post-emergence) were dusted with carborundum (320 grit powder, Fisher Scientific, USA) before mechanical inoculation with virus-bearing sap (ca. 1:10 ratio g tissue/mL H_2_O) [37]. 

### 2.2. Phylogenetic Analysis

Protein sequences derived from three RDR clade members (RDR1, RDR2 and RDR6) from eleven plant species (accession numbers in Appendix A) were aligned with a modified MAFFT program which allows adjustment of several parameters [38]. CmRDR1c1 and CmRDR1c2 were Sanger sequenced for this study, using primers from the sequences of the cucumber analogues and the Melonomics database (https://melonomics.net (accessed on 28 February 2020)) (Appendix A). All other genes were accessed from public databases. Multiple sequence alignment was performed globally for all pairs with a maximum 1000 iterations. The robustness of the multiple sequence alignment was estimated with GUIDANCE2 [39] to be 0.971, which was slightly more robust than the local pair option. The phylogenetic tree was constructed using a likelihood method by running the ‘phyml’ program [40] with the JTT substitution model. The robustness and confidence of the tree nodes was estimated as a percentage of 1000 bootstrap resampling replicates.

### 2.3. Genome Mapping and Expression Analysis by PCR and QPCR 

Total genomic DNA and RNA was extracted from young melon leaves (second or third youngest) using the Dellaporta method [41] and TRI-REAGENT kit (Molecular Research Center, Inc., Cincinnati, OH, USA), respectively. Quantified total RNA was adjusted using a NanoDrop ND1000 spectrophotometer (Thermo Scientific, Wilmington, DE, USA) for equal amounts. First-strand cDNA was synthesized from 2 µg of total RNA using the Verso cDNA Synthesis Kit (Thermo Fisher Scientific, Epsom, UK) including DNase. PCR reactions were performed with the appropriate primers for *CmRDR1* (a, b, c1/c2), *CmRDR2*, *CmRDR6*, and cyclophilin genes based on the Melonomics database (Appendix A). PCR conditions were 2 min at 94 °C, then 30 cycles of 30 s each at 94 °C, 58 °C and 72 °C, and an elongation step of 5 min at 72 °C. Quantitative (Q-PCR) reactions were performed in a volume of 15 µL with 4 µL of diluted cDNA (1/4), 3 pmol of each primer and 7.5 µL of Absolute Q-PCR Sybr Green Mix (Applied Biosystems by Thermo Fisher Scientific, Vilnius, Lithuania). Quantitative analysis was performed using a StepOne (Applied Biosystems) Real-Time PCR system with Q-PCR conditions of 10 min at 95 °C (“hot start”) followed by 40 cycles of 3 s at 94 °C, 15 s at 60 °C and 20 s at 72 °C. The relative expression level of gene accumulation was calculated using the ∆∆Ct method normalized to the housekeeping gene (cyclophilin) using StepOne software version 1.7 (Applied Biosystems) and the samples were compared with one of the samples, which was rated as 1 for comparative purposes. The specificity of the primers was tested prior to the Q-PCR analysis by sequencing of the products and melting curve (Appendix A) and non-specific binding was not found.

### 2.4. CRISPR/Cas9 Binary Construct Design

The pRCS binary vector containing Cas9-sgRNA was generated based on 35S:Cas9-AtU6:sgRNA-eIF4E [42]. The sgRNA-eIF4E was replaced by a sgRNA-RDR1c sequence using the appropriate primers (Appendix A), as per [42].

### 2.5. Agrobacterium-Mediated Transformation

A Vedrantais melon genotype was used for *Agrobacterium-tumefaciens*-mediated transformation according to [37] with a number of changes: leaves were surface sterilized for 15 min in sodium hypochlorite (0.5% active chlorine) solution. Kanamycin (150 mg/L) was added to the selection medium. Rooting medium was composed of MS medium supplemented with indole-3-butyric acid (2 mg/L), cefotaxime (500 mg/L) and kanamycin (150 mg/L). 

### 2.6. Genotyping and Mutant Verification

Genomic DNA was isolated from T0 transgenic and non-transgenic plants by the method of [41]. The presence of the Cas9/sgRNA transgene in T0 lines was confirmed by PCR using specific Cas9 primers (Appendix A). Transgenic lines were genotyped for indel polymorphisms using primers flanking the *CmRDR1c1* and *CmRDR1c2* target regions (Appendix A) and the PCR products were verified for the genetic changes by sequencing and restriction analysis with *BmgB*I. Heterozygotic T0 mutant plants were self-pollinated and in the T3 generation homozygotic mutant lines harboring *cmrdr1c1* and *cmrdr1c2* were selected by PCR analysis for molecular and biological analysis. 

## 3. Results

### 3.1. CmRDR1 Gene Family Mapping in Melon

The *CmRDR1* gene family in melon was mapped based on the cucumber *CsRDR1* gene (*CsRDR1a*, *b*, *c1* and *c2*) sequences [14]. Four putative *CmRDR1* genes, *CmRDR1a*, *CmRDR1b* and the duplicated *CmRDR1c1* and *CmRDR1c2*, were mapped in melon using the Melonomics Database (Figure 1).

*CmRDR1a* and *CmRDR1b* are close to each other (~15 kbp) on chromosome 10 in a positive and negative orientation, respectively, similar to the situation in cucumber [14]. *CmRDR1c1* and *CmRDR1c2* are duplicated genes located within ~520 kbp of each other on chromosome 9. The exon/intron arrangement of the *CmRDR1a*, *CmRDR1c1* and *CmRDR1c2* genes was similar to that of cucumber, with four exons of about the same sizes (Figure 1) [14]. Sequence alignment between *CmRDR1c1* and *CmRDR1c2* encoding regions exhibited 95% homology and 98% identity on the protein level (Appendix A) and the putative 5′ and 3′ UTR exhibited 78% and 65% homology, respectively. However, the promoter (ca. 3 kb) upstream of the coding regions of *CmRDR1c1* and *CmRDR1c2* did not exhibit any sequence similarity (data not shown). CmRDR1a and CmRDR1c exhibited 70% amino acid homology, similar to the level of homology in cucumber [14]. Importantly, *CmRDR1b*, located on Chr 10, was missing two regions at the 5′-end in the first exon (a: 515 bp) and part of exon II (b: 1041 bp) compared with the cucumber gene (Figure 2). The *CmRDR1b* a and b regions (Figure 2) harbor unknown sequences. Truncated *CmRDR1b* sequences were mapped in 10 cultivars of different melon varieties by PCR genomic analysis (Figure 2), which indicated that the *CmRDR1b* enzyme was not generally functional in melon. 

A phylogenetic analysis of RDR1, RDR2 and RDR6 genes of cucurbit species was performed (Figure 3). The neighbor-joining tree derived from protein sequences resolved into three clades: RDR1, RDR2 and RDR6. The melon RDR1 gene family mapped to three clusters: *CmRDR1a*, *Cmrdr1b,* and *CmRDR1c1* and *CmRDR1c2* together. Melon RDR1 genes clustered with their cucurbit homologs (Figure 3). All four *CmRDR1* genes are closely related to those of cucumber (Figure 3), subsequently related to *Citrullus lanatus* and lastly to *Cucurbita pepo* and *Cucurbita moschata* (Figure 3). 

### 3.2. Expression Level of RDR1 Gene Family in Melon Infected with Different Viruses

The expression level of *CmRDR1* genes were determined first in healthy melon by end-point RT-PCR analysis. *CmRDR1a* was constitutively expressed in leaves similar to the levels of *CmRDR2* and *CmRDR6* (Figure 4). However, the expression levels of both *CmRDR1b* and *CmRDR1c1/2* were below the detection limit (Figure 4). *CmRDR1b* is a truncated gene compared with those of cucumber and probably was either not expressed, due to the 5′ gene deletion, or rapidly destroyed as aberrant mRNA (Figure 4) [43]. 

Melon is susceptible to infection by many viruses; therefore, the relative expression of the *CmRDR1* gene family was tested following infection by viruses of different genera and families (Figure 5). 

Interestingly, although the *CmRDR1c1/2* genes were not expressed in healthy plants, these genes are highly induced by infection with a number of viruses (Figure 5 and Figure 6). The highest expression level of *CmRDR1c1/2* was detected in plants upon CMV infection (340-fold increase) (Figure 5a). Following infection by the ipomovirus CVYV or the tobamovirus CGMMV, *CmRDR1c1/2* expression level increased by 173-fold or 115-fold, respectively (Figure 5a). A moderate increase of *CmRDR1c1/2* gene expression of about 5-fold resulted from potyvirus infection (ZYMV, PRSV and WMV) (Figure 5). The expression level of *CmRDR1c1* and *CmRDR1c2* were measured together as *CmRDR1c*, since they have a similar kinetics and response to virus infections (Figure 5 and Figure 6). 

We have recently shown that the CMV-2b suppressor protein was associated with induction of *CsRDR1c* expression [36]. In light of this, we examined the expression level of *CmRDR1c1* and *CmRDR1c2* separately in melon infected with CMV and the CMV-Δ2b mutant. *CmRDR1c1* and *CmRDR1c2* were each expressed about 10-fold less in CMV-Δ2b infection compared with CMV at 9 and 16 days post inoculation (dpi) (Figure 6a,b). The lower expression levels of *CmRDR1c1* and *CmRDR1c2* correlated with the lower level of CMV-Δ2b accumulation and milder symptoms than observed with CMV (Figure 6c). Exceptionally increased expression of *CmRDR1c1* and *CmRDR1c2* was detected in plant 2b-5 infected with CMV-Δ2b 16 dpi (Figure 6d). Sequencing analysis of CMV-Δ2b in this plant revealed a mutation from ACG to ATG that restored 2b functionality, and as a result, virus accumulation and *CmRDR1c1* and *CmRDR1c2* expression levels increased in 2b-5 compared with other CMV-Δ2b-infected plants (Figure 6d).

### 3.3. Expression of CmRDR2, CmRDR6 and CmRDR1a Genes upon Virus Infection

To understand whether other *CmRDR* genes can complement the loss of *CmRDR1b* function in melon, we examined the expression levels of *CmRDR2*, *CmRDR6* and *CmRDR1a* in the susceptible melon cv. Arava in response to different viruses at an early growth stage. The expression levels of *CmRDR1a* and *CmRDR6* decreased ~2-fold and the level of *CmRDR2* was unchanged in ZYMV-infected plants at 7 dpi (Figure 7). Infection by CVYV slightly decreased the expression level of *CmRDR1a* and *CmRDR2*, but the *CmRDR6* level was unchanged. Infection with CGMMV induced *CmRDR1a* and *CmRDR6* expression level ~2-fold, whereas the *CmRDR2* expression was unchanged. An exceptional effect was observed on CMV infection, in which the *CmRDR1a* expression level was induced 21-fold accompanied by moderate increases in *CmRDR2* and *CmRDR6* of 3- and 5-fold, respectively (Figure 7). 

### 3.4. Evaluation of CmRDR1c (1+2) Contribution to Virus Severity

To study the contribution of the *CsRDR1c1/c2* genes to plant virus defense, knockout mutants of these genes were generated by CRISPR/Cas9. Homozygous *cmrdr1c1* and *cmrdr1c2* T3 mutant plants were generated by self-pollination (Figure 8). The selected *cmrdr1c1/c2* mutant line 117 harboring a 5 nts deletion was used in a virus susceptibility assay (Figure 8c). Phenotypically, the mutant *cmrdr1c1/2* (Line 117) plants developed similarly to the non-mutant WT control plants (Figure 8d) and no effect was observed on leaf structure and fruit set and plant development.

*Cmrdr1c1/c2* mutant plants were inoculated with CMV and ZYMV separately, and symptom expression and virus accumulation were examined. CMV RNA accumulation was similar to WT plants 14 dpi (Figure 9a). *Cmrdr1c1/c2* mutant plants (Line 117) infected with CMV exhibited similar severe symptoms as noted in the WT plants (Figure 9b). However, a significant increase of CMV RNA accumulation was measured in mutant plants 24 and 32 dpi. This is despite the fact that the expression level of *CmRDR1c1/2* in WT is tens of times higher than in *Cmrdr1c1/c2* mutant plants, whereas the expression levels of *CmRDR2* and *CmRDR6* were slightly increased in *cmrdr1c1/2* mutant plants. Mutant *cmrdr1c1/c2* plants infected with ZYMV exhibited similar symptoms to the WT plants and ZYMV RNA accumulated to a similar level as in the WT plants at 14 and 24 dpi (Figure 9a,b). Accordingly, the levels of *CmRDR1c1/2*, *CmRDR2* and *CmRDR6* did not change between mutant and WT due to ZYMV infection. The low level of *CmRDR1c1/2* in mutant plants infected with CMV is probably due to rapid degradation of truncated *CmRDR1c1/2* mRNA (Figure 9c), as shown in eukaryotes with mRNAs that prematurely terminate translation [43].

## 4. Discussion

### 4.1. CmRDR1 Gene Organization

Here we report genomic characterization of the gene family *CmRDR1s* in melon, and gene product expression before and after virus infection. Four *CmRDR1* genes were identified (RDR1a, RDR1b RDR1c1 and RDR1c2) with similar genome organization (intron/exon) to that of cucumber (Figure 1) [14]. Interestingly, the RDR1 gene family in *Cucumis* spp. was homologous to those of other genera (*Cucurbita* and *Citrullus*) in the *Cucurbitaceae* (Figure 3), whereas in most plant species only a single or duplicated RDR1 gene has been described [15].

In melon, *CmRDR1c1* and *CmRDR1c2* genes share high sequence similarity, like the duplicated gene described in cucumber [14]. Likewise, duplication of the *RDR1c* gene was homologous in the three other cucurbits examined (watermelon, squash and pumpkin) (Figure 3). Duplication of *RDR1* genes was characterized in barley (*HvRDR1a* and *HvRDR1b*) [16] and potato (*StRDR1a* and *StRDRb*) [15]. 

The close phylogenetic relationship of the *RDR1* gene family in the five cucurbit species (Figure 3), suggests that the four *RDR1* genes were duplicated before the separation of the ancestral cucurbits into different taxa. Therefore, the unique cucurbit *RDR1* gene family may indicate an additional function or interaction between the RDR1 family members in plant defense. In melon, the *CmRDR1* gene family mapped to two chromosomes (9 and 10), whereas in cucumber the *CmRDR1* gene family was located on chromosome 5 [14], which may reflect a cucumber chromosome fusion from a progenitor species [44]. In plants, the region ~3 kbp upstream to *RDR1* has been characterized as a regulatory region [22,45,46]. Therefore, the different sequences (~3 kbp) upstream of the *CmRDR1a*, *b*, *c1* and *c2* genes may imply differential regulation of each RDR1 gene. 

### 4.2. Characterization of Truncated CmRDR1b Gene in Melon Genome

We show here that the *CmRDR1b* gene from melon accessions worldwide was truncated (*cmrdr1b*), and the transcript undetectable due to the two missing regions, compared with the *CsRDR1b* gene in cucumber [14]. We therefore propose that the truncated *CmRDR1b* in melon was mutated in an ancestor during genome duplication and transposon amplification associated with retrotransposon elements in this region [47].

### 4.3. Kinetics of RDR1a, b, c1+c2 Gene Expression Response to Virus Infection

Unique *CmRDR1a, b, c1+c2* expression levels were determined in melon. *CmRDR1a*, which shared the highest homology to *RDR1* from other species, was continuously expressed in healthy leaves in cucumber and other hosts [14,15]. Expression of the truncated *cmrdr1b* was not observed in melon (Figure 1 and Figure 3). In cucumber, the high expression level of *CsRDR1b* in healthy plants was characterized as a marker for broad virus resistance [14] and probably the susceptibility of melon to many viruses may result from malfunction of CmRDR1b. In healthy melon plants the expression level of *CmRDR1c1* and *CmRDR1c2* were below the detection level (Figure 3) similar to *CsRDR1c1* and *CsRDR1c2* in cucumber [14], which may indicate that its regulation was due to virus infection. It is important to note that this unique response of *RDR1c1* and *RDR1c2* was demonstrated only in the genus *Cucumis*, as in other host families *RDR1* is continuously expressed in healthy plants [15,16].

In CMV-infected melon, *CmRDR1a* gene expression was induced 21-fold, but not by *Potyviridae* (ZYMV and CVYV) and a tobamovirus (CGMMV) (Figure 5 and Figure 7). Likewise, in cucumber protoplasts, transfection with CMV increased *CsRDR1a* expression level 29-fold and exogenous application of *CsRDR1a* reduced CMV accumulation dramatically [36], probably due to 2b induction [36]. However, the involvement of CmRDR1a in defense against potyviruses needs additional study. 

The duplicated *CmRDR1c1* and *CmRDR1c2* genes share high sequence similarity and have similar expression levels following virus infection (Figure 6), although the putative promoters upstream of the genes lack homology. Expression of both genes was induced by virus infection strongly and differentially, depending on virus genus. Whereas potyviruses caused gene induction by 4- to 6-fold, CMV, CVYV and CGMMV caused induction by >100-fold (Figure 5). CMV infection induced *CmRDR1c1+2* expression by ~300 fold, similar to *CsRDR1c* in cucumber cv. Beit-Alfa [14]. However, CMV with a deleted 2b gene induced significantly lower *CmRDR1c1+2* transcript accumulation (about 10 to 15-fold lower) (Figure 5), which may indicate that *CmRDR1c1+2* gene expression was regulated by CMV level. Alternatively, the 2b protein can enter the nucleus [22] and can elicit PAMP-triggered immunity [36,48]. In addition, in melon infected with the tobamovirus CGMMV, *CmRDR1c1+2* expression was induced ~100 fold, probably because of the high level of CGMMV-CP accumulation, an oxidative burst inducer as demonstrated in tobacco by tobacco mosaic virus [49] or via tobacco mosaic virus dsRNA inducer molecules [50]. Interestingly, CVYV (Ipomovirus; *Potyviridae*) induced *CmRDR1c1+2* expression ~20-fold higher than the potyviruses ZYMV, PRSV and WMV during the infection of cucumber [14]. This change may reflect the difference between the suppressors of these genera (P1b of CVYV versus the potyvirus HCPro) [51]. Notably, expression of other RDR genes (*CmRDR2* and *CmRDR6*) was induced in melon by CMV infection, but not by potyviruses and tobamovirus, which suggests that the suppressor 2b of CMV may play a role in PAMP-triggered immunity [36].

### 4.4. The Sensitivity of cmrdr1c1+2 Mutant Plants to Virus Infection

Expression of *CmRDR1c1+2* was rapidly induced to a high level by various viruses (Figure 5), an important role in virus-infected plants. Therefore, *cmrdr1c1+2* mutant melon lines were generated to test this role. Mutant lines did not differ in plant growth and fruit development from the WT cultivar. Likewise, ZYMV-infected *cmrdr1c1+2* mutant plants showed similar severe symptoms to the WT plants and virus accumulation did not differ between mutant and WT plants. Similarly in potato, reduction of *StRDR1* expression level by RNAi did not enhance susceptibility to PVY [15]. In addition, no differences of *CmRDR2* and *CmRDR6* expression level were observed between mutant and WT plants. These data suggest that in melon the *CmRDR1c1+2* genes were not involved with potyvirus defense in our experimental conditions. However, we cannot rule out that under different conditions of abiotic or biotic stresses these genes may be important in protection against potyviruses. In contrast to potyvirus, *cmrdr1c1+2* mutant plants infected with CMV showed slightly greater severity compared with WT plants; CMV accumulation was significantly increased 24 and 32 dpi. In addition, *CmRDR1c1+2* were highly expressed in WT plants compared with very low levels of expression in mutant plants, suggesting rapid degradation of aberrant *RDR1c1* and *c2* transcripts of the nonfunctional proteins [43]. These data suggest that *CmRDR1c1* and *CmRDR1c2* are unique genes associated with defense against CMV infection specifically, and the virus–host interaction in cucumber between the inducer 2b and RDR1c1+c2 probably also occurs in melon [36]. Such virus-specific protection has not been reported.

### 4.5. Virus Susceptibility in Melon

Melon is a susceptible host to many viruses due to limited genetic resistances. In addition, the virus recovery phenomenon is less common in melon than cucumber. We show that high expression levels of the unique *CsRDR1b* resistance gene in cucumber is associated with increased and broad virus resistance [14]. We, therefore, suggest that susceptibility of melon to many viruses is a result of a non-functional *CmRDR1b* and repair of this gene using advanced technologies such as CRISPR/Cas9 may significantly improve melon resistance to a range of pathogenic viruses.

## Figures and Tables

**Figure 1 plants-11-01795-f001:**
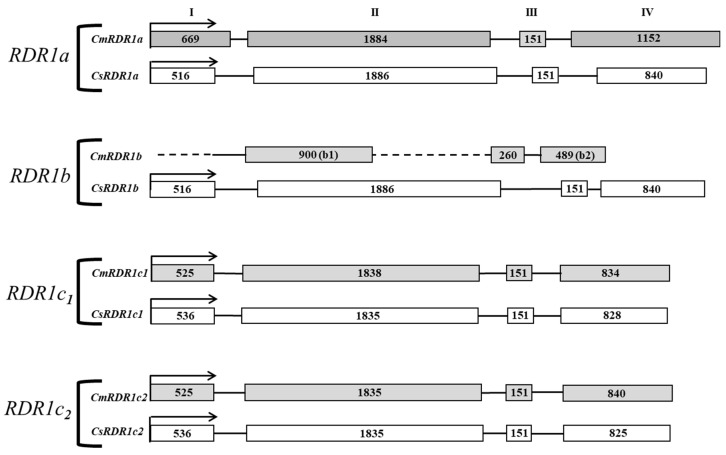
Genome maps of RDR1a, RDR1b, RDR1c1 and RDR1c2 genes in melon (*C. melo*—Cm) and cucumber (*C. sativus*—Cs). Boxes represent exons and lines indicate introns. The dashed lines represent unknown sequences. Genome maps were based on the relevant databases (https://www.melonomics.net/melonomics.html and http://cucurbitgenomics.org (accessed on 28 February 2020)). The numbers indicate exon size (nucleotides) and the start codons (ATG) are represented by an arrow. Exon numbers are marked in Roman numerals (I–IV).

**Figure 2 plants-11-01795-f002:**
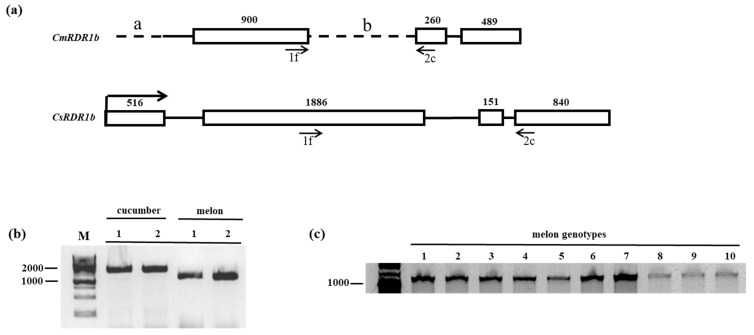
Genome analysis of truncated *CmRDR1b* in melon genotypes. (**a**) Schematic representation of truncated *CmRDR1b* in melon compared with *CsRDR1b* gene in cucumber. Boxes represent exons and lines indicate introns. Dashed lines indicated non-coding sequences (a and b) and arrows show primer sites used for RDR1b size analysis by PCR amplification. (**b**) RDR1b size analysis performed by PCR from cucumber 1928 bp (cv. Ilan) and melon 1613 bp (cv. Arava). (**c**) PCR size analysis of RDR1b genome of 10 different melon varieties (listed in Appendix A). DNA samples were extracted from the third leaf and amplified PCR fragments were separated on 1% agarose gels and visualized by ethidium bromide staining. Size markers (M) (bp) displayed on the left side of the gels.

**Figure 3 plants-11-01795-f003:**
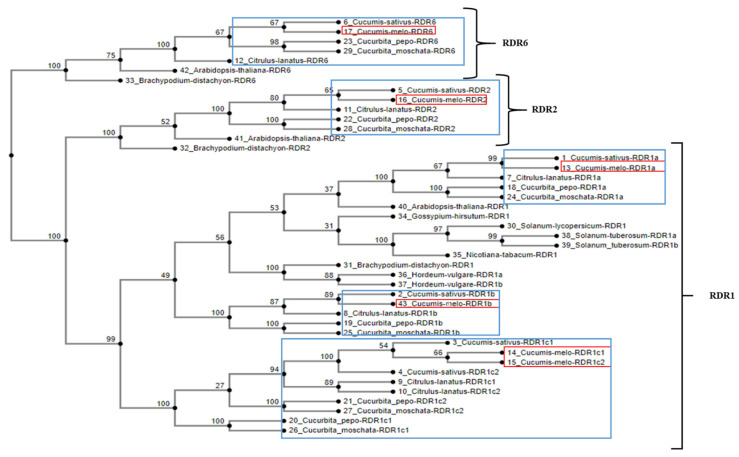
Phylogenetic tree of RDR1, RDR2 and RDR6 gene families of some cucurbit and several other plant species. A phylogenetic tree of five cucurbit species and seven other species was constructed using the maximum likelihood method implemented in ‘phyml’. The analysis separated the genes into three distinct clades: RDR1, RDR2 and RDR6. Values left of the internal nodes are the percentage of bootstrap resampling replicates (out of 1000) that support the tree topology. Only bootstrap values of ≥90% and calculated distances are shown. The red and blue boxes represent *C. melo* and *Cucurbitaceae* genes, respectively. The numbers beside each species name link to the accession number of each gene (listed in Appendix A).

**Figure 4 plants-11-01795-f004:**
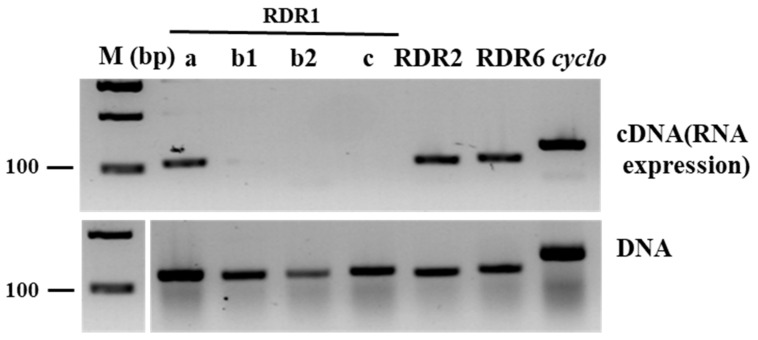
RDR gene expression in healthy melon. Expression of *CmRDR1a*, *CmRDR1b* (b1 and b2), *CmRDR1c1+2* (c), *CmRDR2* and *CmRDR6* genes in healthy melon cv. Arava. RNA expression (**upper panel**) and DNA (**lower panel**) analysis were performed with specific primers. Amplification of *CmRDR1b1* and *CmRDR1b2* fragments was performed using primers flanking exon III. RNA and DNA samples were extracted from the third leaf 10 days post germination and amplified RT-PCR and PCR products were separated on 1% agarose gels and visualized by ethidium bromide staining. Size markers (M) (bp) are shown on the left side of gel.

**Figure 5 plants-11-01795-f005:**
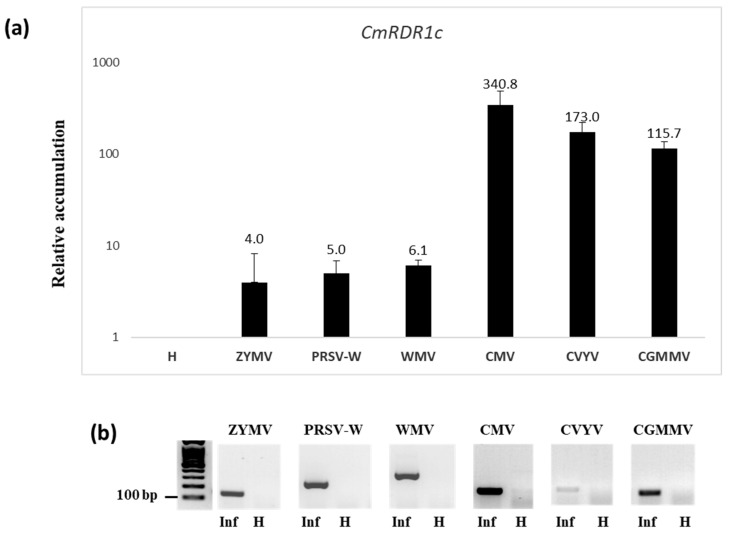
Relative *CmRDR1c1/2* gene expression in melon infected with different viruses. (**a**) Q-PCR analysis of *CmRDR1c1/2* genes expression in ZYMV-, PRSV-, WMV-, CMV-, CVYV- and CGMMV-infected leaves 7 days post infection. The relative levels of virus accumulation were calculated using the ∆∆Ct method normalized to the cyclophilin reference gene using StepOne software version 1.7, and the samples were compared with one rated for comparative purposes as 1 (here one of the ZYMV samples). The mean of each sample is shown with standard error from two leaf disks pooled from each plant, from two plants. (**b**) Virus infection (lower panel) was confirmed by RT-PCR for each virus in healthy (H) and infected (Inf) plants. cDNA samples were separated on 1.5% agarose gels and visualized by ethidium bromide staining. Size markers (bp) are shown on the left side of gel.

**Figure 6 plants-11-01795-f006:**
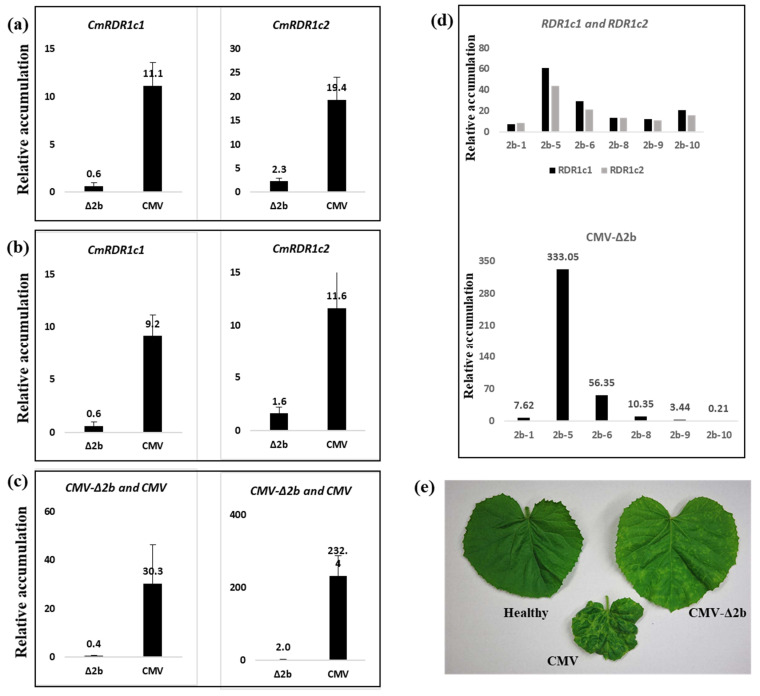
*CmRDR1c1* and *CmRDR1c2* gene expression in melon infected with CMV-Δ2b and CMV. Expression levels of *CmRDR1c1* and *CmRDR1c2* genes were determined by Q-PCR from leaves infected with CMV-Δ2b and CMV at 9 (**a**) and 16 (**b**) dpi. (**c**) CMV-Δ2b and CMV accumulation in infected plants. (**d**) *CmRDR1c1* and *CmRDR1c2* gene expression in melon infected with CMV-Δ2b at 16 dpi. Q-PCR analyses of relative levels of gene accumulation (in (**a**–**d**)) were calculated using the ∆∆Ct method normalized to the cyclophilin host reference gene using StepOne software version 1.7, and then the samples were compared with one infected with CMV-∆2b, rated for comparative purposes as 1. For each CMV-Δ2b sample, two leaf disks were pooled per plant, and the mean of 5 plants with standard error is reported. For CMV infection, 3 plants were tested separately (two leaf disks pooled per plant), and the mean and standard error reported. (**e**) Leaf symptoms of healthy melon and those infected with CMV and CMV-Δ2b at 9 dpi.

**Figure 7 plants-11-01795-f007:**
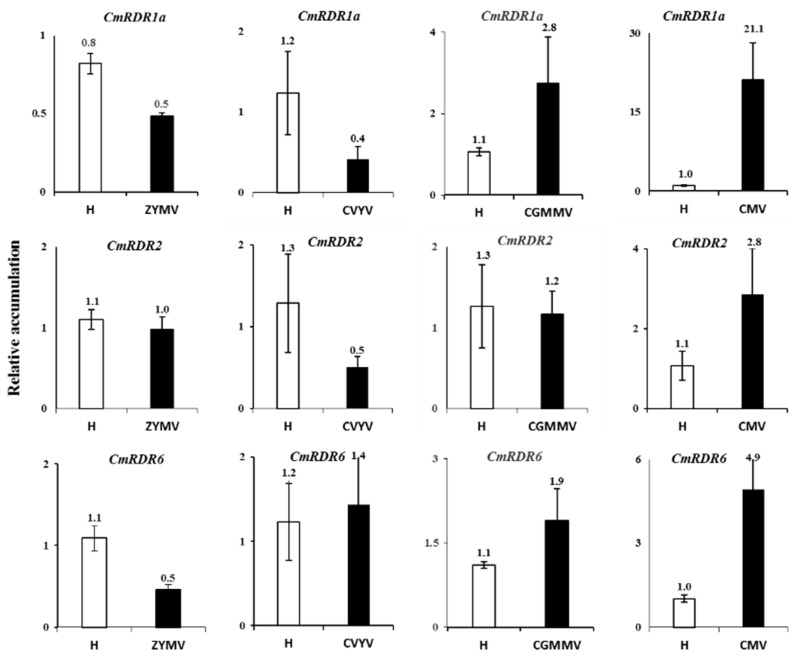
Expression levels of *CmRDR1a*, *CmRDR2* and *CmRDR6* in melon infected with different viruses. Q-PCR analysis of *CmRDR1a*, *CmRDR2* and *CmRDR6* expression in melon infected with ZYMV, CVYV, CGMMV and CMV. White bars represent healthy (H) plants and black bars represent virus infected plants at 7 dpi. Each sample is the mean of pools of two leaf disks per plant from each of 4 plants, with standard error. Relative accumulation was calculated using cyclophilin expression level as a host reference gene.

**Figure 8 plants-11-01795-f008:**
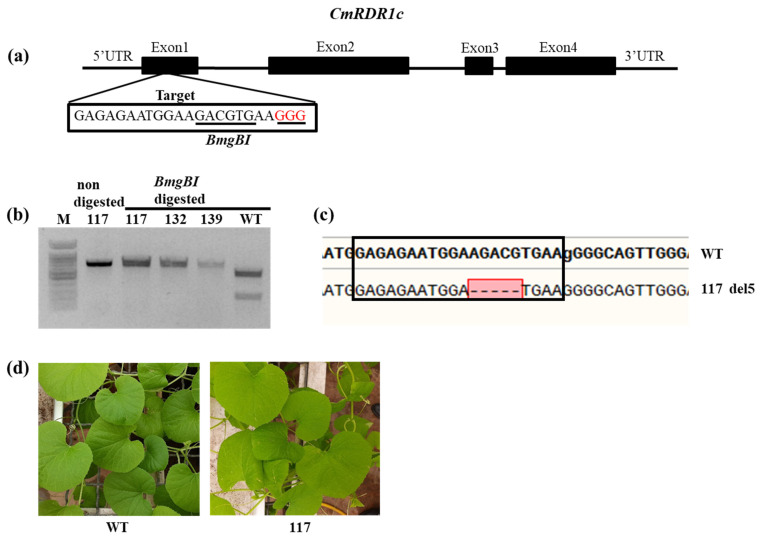
Knockout of *CmRDR1c1/2* genes by CRISPR/Cas9. (**a**) Schematic representation of the *CmRDR1c1/2* genomic maps and the sgRNA target site. (**b**) Restriction analysis for selected mutant lines following digestion of genomic DNA by *BmgB*I, separated on 1% agarose gel. (**c**) Sequence analysis of Line 117 at the sgRNA target (deletion of 5 nts marked by red boxed dashed line). (**d**) Melon phenotype of mutant line 117 compared with WT 14 days post germination.

**Figure 9 plants-11-01795-f009:**
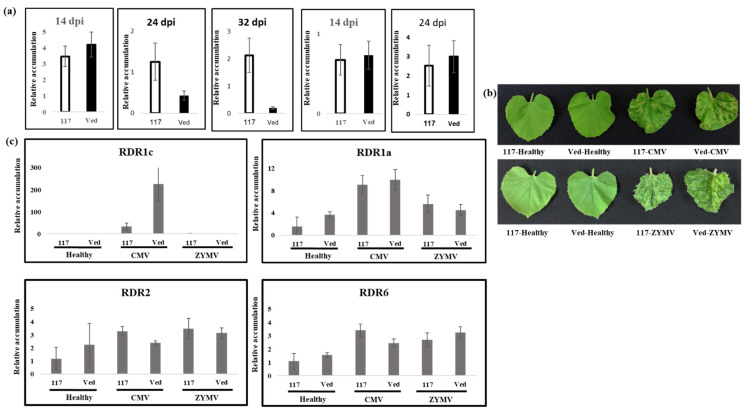
Relative CMV and ZYMV accumulation and expression of *CmRDR* genes in *cmrdr1c1/c2* mutant plants. (**a**) Q-PCR analysis of CMV and ZYMV RNA accumulation in infected mutants (117-white bar) and Vedrantais WT control (Ved-black bar) at different days post inoculation (dpi). (**b**) Disease symptoms in *cmrdr1c1/c2* mutants (117) and Vedrantais WT plants at 14 dpi. (**c**) Q-PCR analysis of *CmRDR1c*, *CmRDR1a*, *CmRDR2* and *CmRDR6* gene expression in mutant (117) and Vedrantais (Ved) plants infected with CMV and ZYMV at 10 dpi. Each result is the mean of pools of two leaf disks per plant from each of 5 plants, with standard error. Cyclophilin was used as an internal host reference for gene expression.

## Data Availability

The data presented in this study are available on request from the corresponding author.

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
