# Peer review of "Analysis of the RNA-Dependent RNA Polymerase 1 (RDR1) Gene Family in Melon"

_plants, 2022, doi:10.3390/plants11141795_

Round 1
Reviewer 1 Report
The manuscript entitled "Analysis of the RNA-dependent RNA polymerase 1 (RDR1) gene family in melon" described the expression pattern and phylogenetic relationship of different RDR genes and their functions against different viruses. The reuslts are interesting and will be useful for screening new antiviral genes.
Here are some comments:
1, the expression of CmRDR1c1/c2 is not detectable in healthy plants, but increased at least 115 fold upon virus infection, this data is not reliable.
2, in the methods section, the author said that the bootstrap resampling replicates is 1000, but in the result section, it said 100, please revise it.
3, line 144, please check the spell of E1f4E.
4, figure 2c, the author did not mark the size of the corresponding marker bands.
5, figure 3, the title need to revise, the tree includes not only cucurbit plants but also other species.
6, figure 8c, it is better to provide the sequencing chromatograms.
7, line 366-369, this statement is not convincing. Additional evidence is needed.
8, the references information need to be revised accordingly. E.g. ref 4 , ref 5, ref 19, ref 20.....
Author Response
- The expression of CmRDR1c1/c2 is not detectable in healthy plants, but increased at least 115 fold upon virus infection, this data is not reliable.
Answer: The reviewer is correct, as we have not explained fully what we did. This has now been clarified in the Materials and Methods at the end of Section 2.3 (lines 143-146), and additionally in the captions to the relevant Figs. 5 and 6.
2. In the methods section, the author said that the bootstrap resampling replicates is 1000, but in the result section, it said 100, please revise it.
Line 231 – Corrected "bootstrap resampling replicates (out of 1000)"
3. Line 144, please check the spell of E1f4E.
Corrected: AtU6:sgRNA-eIF4E.
4, Figure 2c, the author did not mark the size of the corresponding marker bands.
Corrected: The sizes of the marker bands were added to figure 2.
5, Figure 3, the title need to revise, the tree includes not only cucurbit plants but also other species.
Answer: the title was corrected: "Phylogenetic tree of RDR1, RDR2 and RDR6 gene families of some cucurbit and other plant species."
6, Figure 8c, it is better to provide the sequencing chromatograms.
Answer: Since the mutation contain a 5 nts deletion it is not necessarily to show the chromatography.
7, Line 366-369, this statement is not convincing. Additional evidence is needed.
Answer: We observed low level of cmrdr1c1/c2 mRNA in CMV-infected plants. It has been shown that non-translated mRNA triggered degradation of non-translated prematurely terminated translation. We therefore assume (and did not prove) that such a phenomenon occurred in cmrdr1c1/c2 mutant melon line 117.
Answer: The sentence was corrected: "The low level of CmRDR1c1/2 in mutant plants infected by CMV is probably due to rapid degradation of truncated CmRDR1c1/2 mRNA (Figure 9c) as was shown in eukaryotes with mRNAs that prematurely terminate translation. [39]."
8, The references information need to be revised accordingly. E.g. ref 4 , ref 5, ref 19, ref 20.....
Answer: It is unclear to us what the reviewer intends by this remark.
Reviewer 2 Report
The manuscript Plants (ISSN 2223-7747) titled: Analysis of the RNA-dependent RNA polymerase 1 (RDR1) gene family in melon concerns a study on RNA-dependent RNA polymerase 1 role in plant defense against plant viruses, as CMV, and other potyviradae members species.
The argument is interesting. Nevertheless, the manuscript shows some critical feature that in this state cannot be considered for publication.
The introduction should be implemented, the information provided is fragmented and allow to follow the flow of thoughts that lead to the scope of the study. In addition is too short and the last paragraph are results.
The most critical part under my point of view is the material and methods section. Is not clear the flow, thus I suggest producing a flow chart. In addition, this section is confusing and not well written and is not clear to understand if the sequences were obtained in the frame of this study or not. The phylogenetic study is performed on RDR1, RDR2 and RDR6 while the study is focused only in 1, without a justification in the text. The methodology is not clear and primers used not validated.
The discussion is another critical point as results are not sufficient to support the final deduction. The final conclusion in fact assume the lacking of analogue of CsRDR1b in melon explain its susceptibility to plant viruses. in my opinion this is a strong and absolute assessment, that should be strongly provided.

Author Response
English language and style - Extensive editing of English language and style required
Answer: Dr. Victor Gaba, a co-author and native English speaker has edited the ms.
Comments and Suggestions for Authors
Introduction: The introduction should be implemented, the information provided is fragmented and allow to follow the flow of thoughts that lead to the scope of the study. In addition is too short and the last paragraph are results.
Answer: We think that introduction covers the needed details and presents the important data for understanding the paper, as it should be presented. We have also tried to avoid being over lengthy, as the subject matter is well known. It is common to write a short summary of the research results in a sentence or two in the last section of the introduction.
The most critical part under my point of view is the material and methods section. Is not clear the flow, thus I suggest producing a flow chart. In addition, this section is confusing and not well written and is not clear to understand if the sequences were obtained in the frame of this study or not.
We change all the comments noted in the text as follow:
Introduction:
1. Line 55 – Reviewer note Overexpression??
Answer: Yes, this was the finding - probably because the transgene RDR1 activated the gene silencing process.
2. Line 63- Reviewer: mir144 ??
Answer: This is correct: mir144 controls MIKC(C)-type MADS box proteins that repress RDR1 transcription.
3. Lines 64-67. Reviewer: ??
Answer: This sentence was corrected.
4. Line 81: Reviewer: it should be CmRDR1a, CmRDR1b, CmRDR1c1, CmRDR1c2 Answer: We notice "three functional …." CmRDR1b is not functional, therefore the reviewer comment is not correct.
5. Line 83-88: Reviewer: these are results
Answer: It is common to write a short summary of the research results in a sentence or two at the end of the Introduction.
6. Reviewer: this section is not clear. it is not easy to understand how the gene sequences are obtained.
Answer: We have now added near the start of section 2.2 the statement "CmRDR1c1 and CmRDR1c2 were Sanger sequenced for this study, using primers obtained from the sequences of the cucumber analogues and the melonomics database (Table S3)". All other genes were accessed from public database. This information has also been added to the caption of Table S3.
7. Line 108: Reviewer: if I understand well the material for the mechiancal inoculation was obtained from fresh tissues grinding in water? why not a buffer to stabilize the virion integrity:?
Answer: We use water instead of buffer because we found previously that sap-inoculation with H2O did not reduce the efficiency of infectivity.
8. Line 112: Reviewer: why were also included RDR2 and RDR6? is not clear. should be justified.
Answer: The inclusion of RDR2 and RDR6 was as outgroups for the tree-production process, and also gives a broader picture of this group of genes.
9. Line 129, Reviewer: nucleic acid or only RNA? the DNA was not quantified? information about the 230/260 260/280 ratios?
Answer: We purified by the kit only RNA and the Nanodrop data calculated the quantity and quality of the RNA.
10. Line 133, Reviewer: the primer was designed for this study or already published? in the table is not reported any reference, thus the idea is that these primes were designed in this stuudy. No data about the primer efficiences in comparison to cylcophilin reference gene.
Answer: All the primers for the RDRs genes of melon were designed for the first time. The Q-PCR that presented in the figures were calculated using cyclophilin as an reference gene. It is not accurate to compare housekeeping gene with RDRs induced due to biotic and abiotic stress.
11. Line 136, Reviewer: from the information reported it is assumed that syber green chemicals was used but no information is included concerning the melting curve to ascertain the absence of aspecific peaks
Answer: The specificity of each primer were evaluated by sequencing and melting curve and we do not have any non-specific amplification.
Lines 143-146: We added " and the samples were compared to one rated for comparative purposes. The specificity of the primers were tested prior to the Q-PCR analysis by sequencing and melting curve, and no non-specific binding was found."
12. Line 121: Why was used JTT model? how do you choose this one?
Answer: This is program used a simple algorithm which we have used before (ref) and was cited by 5,669 papers.
13. The phylogenetic study is performed on RDR1, RDR2 and RDR6 while the study is focused only in 1, without a justification in the text.
Answer: RDRs genes have a similar enzymatic polymerase activity in the synthesis RNA on a RNA template. These genes are associated with plant regulation and biotic stress including virus infection. In melon the CmRDR1b gene is not functional and therefore it is important show the phylogenetic tree and to evaluate if RDR6 and RDR2 compensate for RDR1b loss (see Fig. 7). Additionally, these related genes serve as outgroups for the RDR1 sequences for the tree-production process, and give a broader picture of this group of genes in the Cucurbitaceae.
14. The methodology is not clear and primers used not validated.
Answer: we clarified the methodology in the text as the reviewer requested. The primers were all used and validated as described in the text.
15. Line 243, Reviewer: end point or quantitative PCR?
Answer : corrected, The "end-point" was inserted into the text
16. The discussion is another critical point as results are not sufficient to support the final deduction. The final conclusion in fact assume the lacking of analogue of CsRDR1b in melon explain its susceptibility to plant viruses. in my opinion this is a strong and absolute assessment, that should be strongly provided.
Answer: Melon and cucumber belong to the Cucumis genus and the four RDR1 genes were duplicated before the separation of the ancestral cucurbits into different taxa. In addition, there is a very high genomic homology between melon and cucumber. In our case, RDR1a and RDR1c have similar gene organizations and share 97%-95% homology. High expression levels of the unique CsRDR1b gene in cucumber genotypes is associated with increased and broad virus resistance. Based on cucumber we therefore assume (but have not proved) that the non-functional CmRDR1b gene may well be associated with virus susceptibility in melon. In a future study we need to prove this by overexpressing a corrected RDR1b gene in melon.
Reviewer 3 Report
the paper which was titled Analysis of the RNA-dependent RNA polymerase 1 (RDR1) 3 gene family in melon is a good paper and written well, few corrections needed
1- add abbreviations of all words the first time appear
2- increased the resolution of figure 3
3- add conclusion
Author Response
1. add abbreviations of all words the first time appear
Answer: All abbreviations have been defined at the point of first use.
2. increased the resolution of figure 3
Answer: Corrected.
3. add conclusion
Answer: We do not feel that a formal Conclusion section is necessary.
Round 2
Reviewer 2 Report
English language and style - Extensive editing of English language and style required
Answer: Dr. Victor Gaba, a co-author and native English speaker has edited the ms.
Comments and Suggestions for Authors
Introduction: The introduction should be implemented, the information provided is fragmented and allow to follow the flow of thoughts that lead to the scope of the study. In addition is too short and the last paragraph are results.
Answer: We think that introduction covers the needed details and presents the important data for understanding the paper, as it should be presented. We have also tried to avoid being over lengthy, as the subject matter is well known. It is common to write a short summary of the research results in a sentence or two in the last section of the introduction.
R: I’m still on my opinion that introduction is too short and not leading to the aim of the study
The most critical part under my point of view is the material and methods section. Is not clear the flow, thus I suggest producing a flow chart. In addition, this section is confusing and not well written and is not clear to understand if the sequences were obtained in the frame of this study or not.
We change all the comments noted in the text as follow:
Introduction:
1. Line 55 – Reviewer note Overexpression??
Answer: Yes, this was the finding - probably because the transgene RDR1 activated the gene silencing process.
R: OK
2. Line 63- Reviewer: mir144 ??
Answer: This is correct: mir144 controls MIKC(C)-type MADS box proteins that repress RDR1 transcription.
R: OK
3. Lines 64-67. Reviewer: ??
Answer: This sentence was corrected.
R: please rephrase
4. Line 81: Reviewer: it should be CmRDR1a, CmRDR1b, CmRDR1c1, CmRDR1c2 Answer: We notice "three functional …." CmRDR1b is not functional, therefore the reviewer comment is not correct.
R: the comment is correct as in the introduction usually is reported the aim and not the results. The aim of your study was to characterized the four genes, then you found that one is not functional, and this part should be included in results and discussion section
5. Line 83-88: Reviewer: these are results
Answer: It is common to write a short summary of the research results in a sentence or two at the end of the Introduction.
R: I completely disagree, the introduction is an introduction where at the end of the paragraph is reported the objective of the study, never results. The results are reported in the results section.
6. Reviewer: this section is not clear. it is not easy to understand how the gene sequences are obtained.
Answer: We have now added near the start of section 2.2 the statement "CmRDR1c1 and CmRDR1c2 were Sanger sequenced for this study, using primers obtained from the sequences of the cucumber analogues and the melonomics database (Table S3)". All other genes were accessed from public database. This information has also been added to the caption of Table S3.
R: OK
7. Line 108: Reviewer: if I understand well the material for the mechiancal inoculation was obtained from fresh tissues grinding in water? why not a buffer to stabilize the virion integrity:?
Answer: We use water instead of buffer because we found previously that sap-inoculation with H2O did not reduce the efficiency of infectivity.
R: please to report data or reference about that. This sentence should be justified
8. Line 112: Reviewer: why were also included RDR2 and RDR6? is not clear. should be justified.
Answer: The inclusion of RDR2 and RDR6 was as outgroups for the tree-production process, and also gives a broader picture of this group of genes.
R: OK
9. Line 129, Reviewer: nucleic acid or only RNA? the DNA was not quantified? information about the 230/260 260/280 ratios?
Answer: We purified by the kit only RNA and the Nanodrop data calculated the quantity and quality of the RNA.
R; You reported in the manuscript: Total genomic DNA and RNA was extracted from young melon leaves (second or third youngest) using the Dellaporta method [36] and TRI-REAGENT kit (Molecular Research Center, Inc., Cincinnati, OH, USA) respectively. Quantified total RNA was adjusted by NanoDrop ND1000 spectrophotometer (Thermo Scientific, Wilmington, DE, USA) for equal amounts. I
Please to specify, not cleare; if not provided a step with DNase, the extracts could have DNA
10. Line 133, Reviewer: the primer was designed for this study or already published? in the table is not reported any reference, thus the idea is that these primes were designed in this stuudy. No data about the primer efficiences in comparison to cylcophilin reference gene.
Answer: All the primers for the RDRs genes of melon were designed for the first time. The Q-PCR that presented in the figures were calculated using cyclophilin as an reference gene. It is not accurate to compare housekeeping gene with RDRs induced due to biotic and abiotic stress.
R: I completely disagree, to verify that the amplification variance is only due to the biotic and abiotic stress the target primer efficiency should be verified to assess to have an efficiency comparable to the housekeeping. Ignoring this point could introduce bias in the amplification variance (https://www.ncbi.nlm.nih.gov/pmc/articles/PMC4280562/)
11. Line 136, Reviewer: from the information reported it is assumed that syber green chemicals was used but no information is included concerning the melting curve to ascertain the absence of aspecific peaks
Answer: The specificity of each primer were evaluated by sequencing and melting curve and we do not have any non-specific amplification.
Lines 143-146: We added " and the samples were compared to one rated for comparative purposes. The specificity of the primers were tested prior to the Q-PCR analysis by sequencing and melting curve, and no non-specific binding was found."
R: include the melting curves as supplementary material
12. Line 121: Why was used JTT model? how do you choose this one?
Answer: This is program used a simple algorithm which we have used before (ref) and was cited by 5,669 papers.
R: as could be reported in the 5669 papers or more, every sequence pattern required a different substitution model. Any software performing phylogenetic analysis have a simple command (find the model), to choose the correct model to that specific sequence data set
13. The phylogenetic study is performed on RDR1, RDR2 and RDR6 while the study is focused only in 1, without a justification in the text.
Answer: RDRs genes have a similar enzymatic polymerase activity in the synthesis RNA on a RNA template. These genes are associated with plant regulation and biotic stress including virus infection. In melon the CmRDR1b gene is not functional and therefore it is important show the phylogenetic tree and to evaluate if RDR6 and RDR2 compensate for RDR1b loss (see Fig. 7). Additionally, these related genes serve as outgroups for the RDR1 sequences for the tree-production process, and give a broader picture of this group of genes in the Cucurbitaceae.
R: OK
14. The methodology is not clear and primers used not validated.
Answer: we clarified the methodology in the text as the reviewer requested. The primers were all used and validated as described in the text.
R: efficiency data is still missing
15. Line 243, Reviewer: end point or quantitative PCR?
Answer : corrected, The "end-point" was inserted into the text
16. The discussion is another critical point as results are not sufficient to support the final deduction. The final conclusion in fact assume the lacking of analogue of CsRDR1b in melon explain its susceptibility to plant viruses. in my opinion this is a strong and absolute assessment, that should be strongly provided.
Answer: Melon and cucumber belong to the Cucumis genus and the four RDR1 genes were duplicated before the separation of the ancestral cucurbits into different taxa. In addition, there is a very high genomic homology between melon and cucumber. In our case, RDR1a and RDR1c have similar gene organizations and share 97%-95% homology. High expression levels of the unique CsRDR1b gene in cucumber genotypes is associated with increased and broad virus resistance. Based on cucumber we therefore assume (but have not proved) that the non-functional CmRDR1b gene may well be associated with virus susceptibility in melon. In a future study we need to prove this by overexpressing a corrected RDR1b gene in melon.
R: The conclusion in a paper should be based on the evidence retrieved in the manuscript. There are no data enough to assume that lacking CsRDR1b analogue in melon explain its susceptibility to plant viruses. is scientifically incorrect.